# Assessment of Squalene-Adenosine Nanoparticles in Two Rodent Models of Cardiac Ischemia-Reperfusion

**DOI:** 10.3390/pharmaceutics15071790

**Published:** 2023-06-21

**Authors:** Romain Brusini, Natalie Lan Linh Tran, Catherine Cailleau, Valérie Domergue, Valérie Nicolas, Flavio Dormont, Serge Calet, Caroline Cajot, Albin Jouran, Sinda Lepetre-Mouelhi, Julie Laloy, Patrick Couvreur, Mariana Varna

**Affiliations:** 1Université Paris-Saclay, Institut Galien Paris-Saclay, CNRS UMR 8612, Pole Biologie-Pharmacie-Chimie, Bâtiment Henri Moissan, 6 Rue d’Arsonval, 91400 Orsay, France; rb.romain.brusini@gmail.com (R.B.);; 2Namur Nanosafety Centre, Department of Pharmacy, Namur Research Institute for Life Sciences (NARILIS), University of Namur (UNamur), 5000 Namur, Belgium; 3Université Paris-Saclay, Inserm, CNRS, Ingénierie et Plateformes au Service de l’Innovation Thérapeutique, ANIMEX, 17 Avenue des Sciences, 91400 Orsay, France; 4Université Paris-Saclay, Inserm, CNRS, Ingénierie et Plateformes au Service de l’Innovation Thérapeutique, MIPSIT, 17 Avenue des Sciences, 91400 Orsay, France; 5Holochem, Rue du Moulin de la Canne, 45300 Pithiviers, France; 6Quality Assistance S.A, Technoparc de Thudinie 2, 6536 Thuin, Belgium; caroline.cajot@quality-assistance.be (C.C.); albin.jouran@quality-assistance.be (A.J.)

**Keywords:** SQAd NPs, preclinical model, drug delivery system, cardiac ischemia and reperfusion

## Abstract

Reperfusion injuries after a period of cardiac ischemia are known to lead to pathological modifications or even death. Among the different therapeutic options proposed, adenosine, a small molecule with platelet anti-aggregate and anti-inflammatory properties, has shown encouraging results in clinical trials. However, its clinical use is severely limited because of its very short half-life in the bloodstream. To overcome this limitation, we have proposed a strategy to encapsulate adenosine in squalene-based nanoparticles (NPs), a biocompatible and biodegradable lipid. Thus, the aim of this study was to assess, whether squalene-based nanoparticles loaded with adenosine (SQAd NPs) were cardioprotective in a preclinical cardiac ischemia/reperfusion model. Obtained SQAd NPs were characterized in depth and further evaluated in vitro. The NPs were formulated with a size of about 90 nm and remained stable up to 14 days at both 4 °C and room temperature. Moreover, these NPs did not show any signs of toxicity, neither on HL-1, H9c2 cardiac cell lines, nor on human PBMC and, further retained their inhibitory platelet aggregation properties. In a mouse model with experimental cardiac ischemia-reperfusion, treatment with SQAd NPs showed a reduction of the area at risk, as well as of the infarct area, although not statistically significant. However, we noted a significant reduction of apoptotic cells on cardiac tissue from animals treated with the NPs. Further studies would be interesting to understand how and through which mechanisms these nanoparticles act on cardiac cells.

## 1. Introduction

Over the past few decades, many approaches have been evaluated to reduce the excessive tissue damages that occur after cardiac reperfusion following a period of ischemia. Myocardial ischemia is one of the leading causes of morbidity and mortality worldwide [1]. The only current therapeutic option for myocardial ischemia is coronary reperfusion using either thrombolytic therapy or primary percutaneous coronary intervention. However, reperfusion paradoxically results in injuries, a phenomenon known as “myocardial reperfusion injury” [2].

Adenosine, an endogenous purine nucleoside continuously secreted both intracellularly and extracellularly, was tested in preclinical models to protect against myocardial ischemia reperfusion injury [3,4]. Normal cells produce concentrations of about 300 nM extracellular adenosine; however, in some cases such as inflammation or ischemia, concentrations may reach up to 600–1200 nM. Extracellular adenosine serves as a signaling molecule through one of the four adenosine receptors (A_1_AR, A_2A_AR, A_2B_AR, and A_3_AR) and activates downstream pathways via cyclic adenosine monophosphate (cAMP). Intracellular adenosine can either be converted to inosine by adenosine deaminase or to AMP by the adenosine kinase [5]. Other than those functions, adenosine displays anti-inflammatory and anti-apoptotic properties, since it inhibits neutrophil adhesion to endothelial cells while decreasing cytokines release and radical oxygen species formation [6,7].

An important clinical application for extracellular adenosine signaling is its potent effect as an arterial vasodilator and inhibitor of platelet aggregation. Two formulations of adenosine Adenocard^®^ and Adenoscan^®^ have been approved by the US Food and Drug Administration (FDA) and are currently available on the market [7]. Further, clinical trials showed that adenosine administered in patients with cardiac ischemia either before [8] or after reperfusion [9,10,11] can improve clinical outcome. Although the therapeutic efficacy of adenosine is well recognized in preclinical and clinical assays, an important limitation of adenosine results from its swift metabolization (less than 10 s) and rapid clearance from blood circulation via cellular uptake primarily by erythrocytes and endothelial cells [12]. Thus, high doses must be injected through systemic administration to obtain a protective effect, possibly causing side effects such as suppression of cardiac function, decreased blood pressure and body temperature, hypotension and sedative side effects [10].

To circumvent these limitations, drug delivery nanomedicines (e.g., nanocarriers loaded with an active molecule) may represent an interesting approach [13], some nanomedicines having already received marketing authorization [14]. However, there are currently no drug delivery systems approved for adenosine and only a few studies have been published on the experimental resolution of cardiac ischemia and reperfusion by using for example liposomes [15] or silica nanoparticles [16] as nanocarriers.

We have developed an original strategy to protect adenosine from metabolization, based on the chemical linkage of this nucleoside to squalene, a natural and biocompatible lipid, which triggers the spontaneous self-assembly of the resulting adenosine-squalene (SQAd) bioconjugate into nanoparticles (NPs) [17]. This allows a considerable increase in nanoparticle drug load, while avoiding the so-called “burst release”, which represent key limitations of other nanomedicines using physical rather than chemical encapsulation processes. Interestingly, SQAd nanoparticles (SQAd NPs) have shown an impressive neuroprotective effect in a model of cerebral ischemia/reperfusion [17].

Therefore, the aim of this study was to evaluate if SQAd NPs were also capable of protective effects in a cardiac ischemia/reperfusion model.

## 2. Materials and Methods

### 2.1. Materials

Squalenyl acetic acid and SQAd bioconjugate were provided by Holochem (Pithiviers, France). Adenosine, D-(+)-Glucose (dextrose, Dxt), H9c2 cell line, Dulbecco’s phosphate buffered saline (PBS), Dulbecco’s Modified Eagle’s Medium (DMEM) classical composition with high glucose and basic formulation without glucose, L-glutamine, phenol red, sodium pyruvate and sodium bicarbonate, Claycomb’s Medium, trypsin, penicillin-streptomycin solution, glutamine, norepinephrine, HEPES, 3-(4,5-dimethylthiazol-2-yl)-2,5-diphenyltetrazolium bromide (MTT), Trypan Blue, Evans Blue, 2,3,5-triphenyltetrazolium chloride (TTC), and Bovine Serum Albumin (BSA) were all purchased from Sigma-Aldrich (L’lsle-d’Abeau, Chesnes, France). Foetal Bovine Serum (FBS) was purchased from Life Technologies (Asnières-sur-Seine, France). Absolute ethanol came from VWR Chemicals (Rosny-sous-Bois, France), while dimethyl sulfoxide was from Carlo Erba Reagents (Val-de-Reuil, France). Ultra-pure water was purified using a MilliQ system from Millipore Corporation (Guyancourt, France). CholEsteryl 4,4-Difluoro-5-(4-Methoxyphenyl)-4-Bora-3a,4a-Diaza-s-Indacene-3-Undecanoate (CholEsteryl BODIPY™ 542/563 C11) was purchased from ThermoFisher Scientific (Les Ulis, France). ROTI^®^ Histofix 4% was obtained from Carl Roth (Lagny-sur-Marne, France).

HL-1 Cardiac Muscle Cell Line (HL-1) originated from Sigma-Aldrich (L’lsle-d’Abeau, Chesnes, France). Haematoxylin, eosin and Picrosirius red were purchased from VWR (Rosny-sous-Bois, France).

C57Bl/6J male mice (5–8 weeks old) and Sprague-Dawley male rats (250–350 g) were obtained from Janvier Labs (Le Genest-Saint-Isle, France). All anaesthesia and analgesia materials were kindly provided by our animal house Animex (Paris-Saclay University). The TUNEL Assay Kit—BrdU-Red was obtained from Abcam (Cambridge, England).

### 2.2. Preparation and Characterization of Squalene-Adenosine Nanoparticles

SQAd bioconjugates were synthesized as previously described and nanoparticles (SQAd NPs) were prepared using the nanoprecipitation technique [17]. Briefly, SQAd bioconjugates were dissolved in absolute ethanol at a concentration of 6 mg/mL and 333 µL of the solution were added dropwise into a 5% (*w*/*v*) dextrose (Dxt) solution under strong mechanical stirring. Ethanol was then completely removed by evaporation using a Rotavapor (80–90 rpm, 40 °C, 43 mbar) to obtain a 2 mg/mL aqueous suspension of pure nanoparticles. Fluorescent SQAd NPs were prepared using the same methodology, with the only difference that 1% (wt/wt) of CholEsteryl BODIPY 542/563 C11 was additionally dissolved in the ethanolic phase before addition to the dextrose solution.

SQAd NPs size (hydrodynamic diameter), polydispersity index and surface charge (zeta potential) were assessed by Dynamic Light Scattering (DLS) after preparation, up to 14 days, for stability studies, using a Malvern Zetasizer Nano ZS (Palaiseau, France) (173° scattering angle, 25 °C, Material RI: 1.49, Dispersant Viscosity RI: 1.330, Viscosity 0.8872 cP). For each preparation, the mean diameter resulted from the average of three measurements of 60 s each and the mean zeta potential resulted from the average of three measurements in automatic mode, followed by the application of the Smoluchowski equation.

NPs morphology was evaluated by Cryogenic Transmission Electron Microscopy (cryo-TEM), as follows: drops of NPs suspensions at 2 mg/mL concentration were deposited on electron microscopy grids covered with a holey carbon film (Quantifoil R2/2) previously treated with a plasma glow discharge. Observations were conducted at low temperature (−180 °C) on a JEOL 2010 FEG microscope operated at 200 kV. Images were recorded with a Gatan camera.

### 2.3. Cell Culture

HL-1 cells, an immortalized cardiac muscle cell line derived from mouse atrial cardiomyocyte tumor lineage, were cultured in Claycomb’s medium supplemented with 10% (*v*/*v*) FBS, 2 mM of L-Glutamine, 0.1 mM of Norepinephrin, and 1% (*v*/*v*) of penicillin-streptomycin. Rat cardiomyoblast (H9c2) cells were cultured in complete medium composed of DMEM D6429 with 10% FBS and 1% penicillin–streptomycin. Routinely, cells plated on 75 cm^2^ cell culture flasks were maintained in a humidified incubator at 37 °C (air with 5% CO_2_). Cells were then cultured until a confluence of 70–80% was reached, passaged twice a week and used for experiments between passages 4 and 12.

### 2.4. Cell Viability

Cell viability was assessed by performing a 3-(4,5-dimethylthiazol-2-yl)-2,5-diphenyltetrazolium bromide (MTT, Sigma-Aldrich) colorimetric assay following manufacturer’s recommendations. Briefly, cells were seeded in 96-well plates at 8000 (for HL-1) or 10,000 (for H9c2) cells per well and incubated for 24 h. Cells were then washed with PBS and incubated with different concentrations of nanoparticles ranging from 1 µM to 200 µM for 2, 6 or 24 h. Subsequently, cells were washed with PBS, and incubated for 2 h with medium containing 0.5 mg/mL of MTT. Finally, medium with untransformed MTT was removed and formazan crystals were solubilized by adding 200 µL of dimethyl sulfoxide (DMSO) per well. After 10–15 min of moderate shaking on an orbital shaker, absorbance was measured at 570 nm to determine cell viability, using a Perkin Elmer microplate reader. Each treatment was assessed with 6 replicates, and experiments were performed in triplicate.

### 2.5. In Vitro Cellular Uptake of Fluorescent SQAd NPs

Fluorescent SQAd NPs labelled with CholEsteryl BODIPY 542/563 C11 (SQAd-BP NPs) were used to investigate the cell internalization of the nanoparticles. HL-1 cells (100,000 cells/well) were seeded in 6-well plates on 12 mm glass coverslips. H9c2 cells were seeded in 8-well Ibidi plates at a density of 25,000 cells per well.

Cells were left to adhere and grow in a humidified incubator at 37 °C (air with 5% CO_2_) for 24 h.

Then, the cells were incubated with SQAd-BP NPs (50 µg/mL) diluted in complete culture medium for up to 24 h. Following the incubation step, cells were washed with PBS and fixed with 4% paraformaldehyde (PFA) for 15 min. A 50 mM NH_4_Cl solution in dH_2_O was then added in each well for 15 min, to neutralize residual PFA. Finally, coverslips were mounted with slides and imaged by confocal microscopy with an inverted LSM510 Zeiss confocal microscope (Carl Zeiss, Oberkochen, Germany) using a PlanApochromat 63× objective lens (NA 1.40, oil immersion). SQAd-BP NPs were excited with a Helium-Neon laser (543 nm wavelength). Emitted fluorescence was detected with a 560 nm Long-Pass filter. The pinhole diameter was set at 1.0 Airy Unit giving a 0.8 µm optical slice thickness. Stacks of images were collected every 0.4 µm along the z axis. Numerical images were acquired and analysed with LSM 510 version 3.2.

### 2.6. Toxicological Assessment on Blood from Healthy Donors

#### 2.6.1. Blood Drawn

Blood was obtained for each test, from 3 healthy donors (ethical number B03920096633) who were free from any medication for at least two weeks. All protocols were in accordance with the Declaration of Helsinki and approved by the Medical Ethical Committee of the CHU Dinant-Godinne UCL Namur (Yvoir, Belgium). Blood was collected with a 21-gauge needle via atraumatic antecubital venipuncture in 0.109 M sodium citrate (9:1 *v*/*v*) tubes Venosafe^®^ (Terumo Europe, Leuven, Belgium) for red blood cells (RBCs) and platelet collection and in EDTA tubes Venosafe^®^ (Terumo Europe, Leuven, Belgium) for white blood cells isolation.

#### 2.6.2. Hemolysis Assay

The hemolysis assay was based on the photometric measure of free hemoglobin released in the supernatant due to hemolysis of red blood cells. Triton X-100 (1% (*v*/*v*)) and PBS were respectively chosen as positive and negative control. Each experiment was performed in triplicate. 5 µL of SQAd NPs (stock solution at 1mg/mL), 5 µL of PBS (negative control), or 5 µL of Triton X-100 (positive control) were added to 45 µL of whole blood. The suspension was incubated at room temperature (RT) on an orbital plate shaker (250 rpm) for 3 h. After incubation, samples were centrifuged for 10 min at 2000× *g*. 20 µL of supernatant were dispatched to a 96-well plate. 180 µL of Drabkin’s (Sigma Aldrich, Overijse, Belgium) reagent was added to each well, mixed and left at RT for 15 min. The supernatant was read in a 96-well plate by using a multimode microplate reader SpectraMax iD3 (Molecular Devices, San Jose, CA, USA) at 540 nm. The color intensity was measured at 540 nm and is proportional to the total hemoglobin concentration. The percentage of hemolysis (H (%)) was calculated with optical densities (OD) as followed (Equation (1)):(1)$$H\%=\frac{\left(OD\;550\;nm\;sample-OD\;550\;nm\;negative\;control\right)}{\left(OD\;550\;nm\;positive\;control-OD\;550\;nm\;negative\;control\right)}\times 100$$

Positive and negative controls induced 100% and 0% of lysis, respectively. The results were expressed in percentage as mean ± SD.

#### 2.6.3. Preparation of Human Platelet-Rich Plasma (PRP), Platelet-Poor Plasma (PPP)

Platelet-rich plasma (PRP) was carefully prepared by centrifugation of whole blood at 200× *g* at RT for 10 min. Platelet-poor plasma (PPP) was subsequently obtained by centrifugation at 2000× *g* of the pellet at room temperature for 10 min. The platelet count in PRP was adjusted to 300,000 platelets/µL after dilution with PPP and used immediately after preparation.

#### 2.6.4. Platelet Functional Assay: Light Transmission Aggregometry (LTA)

The impact of SQAd NPs on induced platelet aggregation was investigated using the chronometric aggregometer type 490-2D as previously reported [18,19]. Briefly, 280 µL of PRP at 300,000 platelets/µL were mixed, with respectively 30 µL of adenosine diphosphate (ADP, final concentration: 20 µM, Bio/Data corporation, Horsham, PA, USA), 30 µL of collagen (final concentration: 190 µg/mL, calf skin, Bio/Data corporation, USA) or 6 µL of arachidonic acid (AA, final concentration: 600 µM, Calbiochem, Darmstadt, Germany) and 15 µL of SQAd NPs at final concentration ranging from 100 to 0.1 µg/mL. Inducers alone were also used before each experiment to check the platelet reactivity. PPP was used as reference. Data was collected with the Chronolog two-channel recorders at 405 nm connected to a computer.

#### 2.6.5. Isolation of PBMCs (Peripheral Blood Monocellular Cell)

PBMCs isolation was done as recommended in the Ficoll^®^ notice. 4 mL of whole blood collected in EDTA tubes were twice diluted in PBS and added to 3 mL of Ficoll^®^ reagent before centrifugation at 400× *g* for 35 min. Supernatant containing plasma and platelets was removed. PBMCs were collected and washed with PBS by centrifugation at 400× *g* for 10 min. The cells were collected, suspended in 6 mL of PBS, then centrifuged at 200× *g* for 5 min. The suspension volume was adjusted to obtain 750,000 PBMCs/mL.

#### 2.6.6. Cell Viability (MTS) and Cytotoxicity (LDH) Assays

PBMCs were seeded (150,000 cells/well) in a 96-well plate and incubated for 24 h with SQAd NPs at concentrations ranging from 100 ng/mL to 100 µg/mL. Cell viability and cytotoxicity of SQAd NPs were determined 24 h later using MTS assay (CellTiter 96^®^ AQueous One Solution Cell Proliferation Assay, Promega, WI, USA) or lactate dehydrogenase (LDH) assay (Cytotoxicity Detection KitPLUS, Roche, Basel, Switzerland), according to manufacturer’s instructions. Positive control (Triton X100—1%) and negative control (Dextrose 5%) were equally included.

### 2.7. Animal Care

6 to 8-weeks-old male C57BL/6J mice were purchased from Janvier Labs (Le Genest-Saint-Isle, France) for ischemia and reperfusion studies. 225–250 g Sprague-Dawley rats were purchased from Janvier Labs (Le Genest-Saint-Isle, France) for pharmacokinetics and biodistribution studies. Animals were housed in groups of 5 for mice and groups of 3 for rats and allowed seven days to acclimatize in a standard controlled environment (22 °C ± 1 °C, 60% relative humidity, 12-h light/dark cycles) with food and water available *ad libitum*. All experimental protocols were approved by the Animal Care Committee of the University Paris-Saclay (project n°2018100117547682_v1), in accordance with principles of laboratory animal care and European legislation 2010/63/EU. All efforts were made following the 3R strategies (Reduction, Replacement and Refinement) to reduce animal numbers and minimize their suffering, as defined in the specific agreement.

### 2.8. Mouse Myocardial Ischemia/Reperfusion Model

All surgical procedures of the mouse myocardial Ischemia and Reperfusion (IR) model were performed based on a previously described protocol [20]. Briefly, mice were mildly anaesthetized by intraperitoneal injection of a ketamine/xylazine mix solution (100 mg/kg of ketamine and 10 mg/kg of xylazine) for first sedation and analgesia, followed by full anaesthesia with 3% isoflurane and 0.2 L/min oxygen in an adapted mouse nose mask. Mice remained under anaesthesia until there was a clear loss of paw and tail reflexes. Then, mice were endotracheally intubated and placed on a homeothermic heating pad under mechanical ventilation with a rodent respirator (Harvard Apparatus, Les Ulis, France) supplying 1% isoflurane anaesthesia for the rest of the surgery. Mice were shaved, scrubbed with Vetedine and positioned into a right lateral decubitus position under a dissecting microscope. Left thoracotomy was then performed in the third intercostal space. The thoracic cage was maintained open thanks to chest retractors, the pericardium was gently separated, and Left Anterior Descending (LAD) ligation was performed using a 7-0 silk suture ligated around a piece of PE-10 tubing, 1–2 mm below the left auricle. A second ligation was placed without tying it up for future staining experiments. Complete occlusion of the vessel was confirmed by whitening of the ventricle after ligation. After 30 min of ischemia, the tubing was removed, and the first ligation was cut to allow reperfusion of the heart. Right after reperfusion, intravenous injection of the treatment (15 mg/kg) was performed into the left jugular vein. Subsequently, pneumothorax was evacuated manually, mice were sutured and subcutaneously injected with buprenorphine (0.1 mg/kg body weight) analgesia. Gas anaesthesia was stopped, intubation was removed after the mice began to breathe on their own, and finally allowed to recover in an oxygenated and heated chamber for a few hours, before returning into normal housing conditions for different periods of reperfusion.

### 2.9. Area at Risk (AAR) and Infarct Area (IA) Evaluation

Three or seven days after reperfusion, mice were sacrificed to obtain blood samples and hearts. Briefly, mice were intraperitoneally injected with high doses of pentobarbital (>100 mg/kg). The right femoral vein was exposed, and a maximum amount of blood was collected (generally between 500–800 µL), using heparin-coated syringes. Blood samples were directly put on ice, plasma was separated by centrifugation (2000× *g*, 10 min at 4 °C) and finally stored at −80 °C. After blood was collected, 20 mL of PBS was injected to wash out residual blood. Subsequently, the LAD artery was re-occluded with the suture left in situ, and 5 mL of 0.5% Evans Blue solution was injected to delimitate the Area At Risk (AAR). Hearts were then excised and put into a saturated KCl solution to stop the heart at the diastolic phase; auricles were removed, and the hearts were sliced into 1-mm thick perpendicular cross sections. Heart sections were then incubated with a 1% 2,3,5-triphenyltetrazolium chloride (TTC) solution at 37 °C for 15 min to determine the Infarct Area (IA). After TTC staining, both sides of each section were imaged with a Zeiss microscope, and IA (negative for TTC and Blue Evans staining and appearing pale white), AAR (appearing pink/pale red, without Blue Evans staining) and total left ventricular area (LV) were assessed using ZEN v2.3 software (Zeiss) in a blind manner. The percentage of area at risk was measured as the ratio (AAR/LV) × 100, and the infarct size was calculated as (IA/AAR) ×100. Images were analysed in a blind manner to avoid bias. Sections were kept in 4% paraformaldehyde solution for 24 h (at 4 °C), and finally either frozen at −80 °C or embedded into paraffin.

### 2.10. Histological and Immuno-Histochemical Analyses

Heart tissues were collected and fixed in 4% paraformaldehyde for histology. Fixed tissues were washed, dehydrated, and submitted to paraffin embedding. Paraffin-embedded heart tissues were cut in serial 5 µm-thick cross sections, using a microtome and stained with hematoxylin and eosin (H&E) for histological evaluation of tissue damage, or stained with Picrosirius Red to evaluate the fibrosis. Diseased areas with scar tissue and areas with abnormal modifications such as infiltration of inflammatory cells, apoptotic/necrotic cells, fibrosis were delimited and compared between treatment groups. Other heart sections were directly cut in serial 5 µm-thick frozen cross sections, using a cryostat microtome.

Paraffin 5 µm cut sections were treated with sodium citrate buffer (pH 6.0) for antigen retrieval. Cryofrozen 5 µm sections were rehydrated before their uses for immunohistochemistry. All sections were incubated in a humidified dark chamber at 4 °C overnight with one of the following antibodies: iNOS (GTX130246, 1:100, Euromedex, Souffelweyersheim, France), or CD206 (GTX 42264, 1:100, Euromedex, France). The following day, sections were washed with PBS and incubated for 2 h at RT with the appropriate secondary antibodies conjugated to HRP: anti-rabbit (GTX213110, 1:3000, Euromedex, Souffelweyersheim, France) or anti-rat (GTX26955, 1:3000, Euromedex, Souffelweyersheim, France). Further, the immunostaining was revealed using a diaminobenzidine (DAB) kit (Sigma). For immunofluorescence, corresponding fluorescent secondary antibodies (DyLight 633) were used (1:3000), incubated for 1 h at room temperature and, after rinsing, the sections were further mounted with aqueous mounting media containing DAPI for nuclei staining.

Digital imaging of immunostained sections was performed on 6–10 random fields by confocal microscopy, using a 40× objective and counted.

### 2.11. Evaluation of Apoptosis in Tissue Sections

Apoptosis was evaluated on paraffin sections with DNA fragmentation detection by Terminal deoxynucleotidyl transferase dUTP nick-end labeling (TUNEL) staining, using an in situ BrdU-Red DNA fragmentation assay kit, according to the manufacturer’s recommendations.

Briefly, paraffin sections were deparaffinized and rehydrated through successive incubations in toluene, and then in less and less concentrated ethanolic solutions. After immersing sections in PBS, Proteinase K solution was added for 15 min at room temperature to retrieve antigens. Then, cells were labelled using the manufacturer DNA labelling solution for 60 min at 37 °C. After washing the section slides in PBS, the manufacturer antibody solution comprising anti-BrdU-Red antibody was added for 30 min in the dark at room temperature. DNA counterstaining was finally performed using the manufacturer’s 7-AAD/RNase A staining buffer for 30 min in the dark at room temperature. After last incubations in ddH2O washing solution, coverslips were added, and sections were analysed by fluorescence confocal microscopy with 40× objective (Ex/Em = 488/576 nm (BrU-Red) and Ex/Em = 488/655 nm (7-AAD)). TUNEL-positive nuclei and total nuclei were indicated by red and blue fluorescence, respectively. The number of TUNEL-positive nuclei was counted by examining 6 areas around the ischemic zone on the sections.

### 2.12. Pharmacokinetics and Heart Accumulation of NPs

The pharmacokinetic studies were carried out in rats because high amounts of samples were required for different analysis. The blood samples (250 µL/time point) were taken from rats without cardiac ischemia/reperfusion, at different time points after intravenous administration of 15 mg/kg of SQAd NPs: 5 min, 15 min, 30 min, 1, 2, 6 or 24 h. 250 µL of blood was taken from rats with cardiac ischemia/reperfusion at different time points after intravenous administration of 15 mg/kg of SQAd NPs: 15 min, 1, 2, or 6 h. On these rats a volume of 250 µL of whole blood was also collected one week before ischemia/reperfusion. At the end of experiment, the rats were killed (injection of a lethal doses of pentobarbital), the heart and whole blood were collected. Rapidly, plasma was separated by centrifugation (3000× *g*, 10 min) before freezing at −20 °C. Cardiac tissue was separated into two tubes (ischemic and non-ischemic zones), immediately frozen at −20 °C and sent to Quality Assistance for further analysis. The quantification of adenosine and its metabolites (Inosine and Hypoxanthine) in whole blood and cardiac tissue was performed according to the protocol established by Quality Assistance. The procedure developed involves a protein precipitation step and separation by UPLC under reverse phase conditions. The different products were detected by tandem MS (Mass Spectrometry). The limit of detection was 1 to 500 ng/mL for Adenosine, 5 to 250 ng/mL for Hypoxanthine 5 to 250 ng/mL for Inosine and 5 to 200 ng/mL for SQAd.

### 2.13. Statistical Analyses

Statistical calculations were performed using a Student *t* test. *p*-values of 0.05 or less were considered statistically significant. Data are expressed as mean ± S.E.M.

## 3. Results and Discussion

### 3.1. Preparation and Characterization of SQAd NPs

Squalene-adenosine (SQAd) was synthetized by covalent conjugation of adenosine to squalenyl acetic acid (drug loading 37%) as previously described [17] in order to protect adenosine from quick metabolization.

SQAd nanoparticles were obtained with the nanoprecipitation method, as a monodisperse aqueous suspension, as observed by cryo-TEM (Figure 1A). Nanoparticles stability was evaluated by dynamic light scattering (DLS) in water supplemented with 5% dextrose, at 4 °C and at room temperature (RT). NPs were stable for up to 14 days, displaying a mean hydrodynamic diameter of 92.8 ± 1.03 nm at 4 °C and 94.1 ± 1.72 nm at RT (Figure 1B), a polydispersity index of 0.094 ± 0.017 and 0.109 ± 0.015 respectively, and a negative zeta potential surface charge (Figure 1C).

These results were consistent with those found previously by Dormont et al. [21]. The nanoprecipitation procedure was very accessible without any excipients required and allowed freeze-drying while preserving the nanoparticles’ diameter, zeta potential and supramolecular structure [22]. This represents a significant benefit for the scaling-up and clinical translation, in comparison to the more complex nanomedicines currently in development for the treatment of cardiac diseases. These NPs were previously deeper characterised by using Small-Angle X-ray Scattering (SAXS) and we evidenced an inverse two-dimensional hexagonal phase resulting from the stacking of inverse cylinders constituted of adenosine-squalene bioconjugates [17].

### 3.2. SQAd NPs Cytotoxicity Assessment

The HL-1 cardiomyocyte cell line, derived from mouse atrial cardiomyocyte tumour cells, has been extensively explored as in vitro models of cardiac ischemia/reperfusion (IR) injury with similar drug responses to IR as primary cardiomyocytes [23,24]. This cardiac cell line retains its ability to proliferate and can be passaged repeatedly without reverting to an embryonic phenotype [23]. Thus, HL-1 was chosen as the cell model for evaluation of cell viability, and cell internalization of SQAd NPs.

As shown in Figure 2A, SQAd NPs did not disturb the cellular viability after 2 h, 6 h and 24 h of treatment, neither at low nor at high concentration. Even at 150 µg/mL, no cytotoxicity was noted. Free adenosine and squalenyl acetic acid nanoparticles (SQCOOH NPs) were tested as controls and did not show any cytotoxicity at the equivalent maximum concentration of SQAd NPs either.

We further performed cytotoxicity study on the H9c2, a cardiomyoblast cell line, widely used and described in the literature. The results obtained (Figure 2B) show that H9c2 cells are more sensitive at very early times (i.e., 2 h) after addition of NPs, but only at higher doses (more than 50 µg/mL). This effect is no longer observed at 24 h. A deep study to understand by which pathways NPs enter into the endothelial and cardiomyoblast cells, and the fate of NPs inside the cells would be interesting.

### 3.3. Cellular Uptake of SQAd NPs

To investigate the cellular uptake, Bodipy fluorescent SQAd NPs (50 µg/mL) were incubated with HL-1 or H9c2 cells in complete medium and internalization was monitored by confocal microscopy. For HL-1 cells, the maximum fluorescence intensity was detected at 24 h (Figure 3, upper panel), whereas no fluorescence was visible when the cells were incubated with the free fluorescent probe (Figure 3, lower panel). This suggests that SQAd NPs progressively accumulate in HL-1 cardiac cells over time.

For H9c2 cells a strong fluorescence was detected starting from 5 h with a maximum at 24 h (Figure 4, upper panel). No fluorescence was visible when the cells were incubated with the free fluorescent probe (Figure 4, lower panel).

These results were concordant with our previous work done with H9c2, (cardiomyoblast cell line) and MCEC (Mouse Cardiac Endothelial Cells), where accumulation of fluorescent SQ-based NPs was strongly detected after 18 h of incubation [25]. In a previous work, our group demonstrated that the LDL Receptor was involved in SQAd NPs internalization by HepG2 cells [26]. Further analysis to elucidate the mechanism of uptake by cardiac cells is ongoing.

### 3.4. Toxicological Assessment of SQAd on Peripheral Blood

The hemolytic effect of SQAd NPs was determined on whole blood to mimic clinical conditions more closely. After incubation for 3 h at room temperature, the obtained results show that there was no hemolysis, regardless of the tested concentration of NPs.

To evaluate the anti-aggregating capacities, different concentrations of SQAd NPs were incubated with platelets. The obtained results demonstrated the NPs’ anti-aggregating effect at concentrations greater than 10 µg/mL after treatment of platelets (PRP) with substances known to induce platelet aggregation: ADP (Figure 5A), Collagen (Figure 5B) or Arachidonic acid (Figure 5C). Inhibition of platelet aggregation was not observed when PRP were incubated with the lowest concentrations of SQAd NPs (e.g., 1 µg/mL or 0.1 µg/mL).

To evaluate the impact on blood cells, the PBMC were incubated with different concentrations of SQAd NPs for 24 h to assess both cell membrane integrity, determined by LDH, as well as cell viability by measuring mitochondrial activity (MTS). Figure 5D shows that PBMC incubated with different concentrations of SQAd NPs displayed very low release of LDH, even when incubated at high concentrations (i.e., 100 µg/mL). Similarly, the MTS test (Figure 5E) evidenced very good cell viability under the same incubation conditions. We had previously shown the lack of toxicity when SQAd NPs were administered intravenously in mice [17]. In the present study, the toxicity of the nanoparticles was analyzed, for the first time, on human peripheral blood, an important investigation for further clinical translation.

### 3.5. Area at Risk, Infarct Size and Fibrosis Evaluation

Left anterior descending (LAD) coronary artery ligation in rodent models represents the “gold standard” to simulate I/R events associated with myocardial infarction [27,28]. Here, we performed LAD artery ligation for 30 min followed by 3 or 7 days of reperfusion. Treatment was administered in the jugular vein immediately after reperfusion.

At the end of the treatment, animals were sacrificed, Evans Blue and triphenyltetrazolium chloride (TTC) staining were performed to delimit the infarct area (IA) and the area at risk of infarction (AAR). As shown in Figure 6A, 3 days after treatment, the AAR was smaller in the SQAd NPs treated group (27.4% ± 10%) compared to the free adenosine treated group (41.4 ± 6.5%), as well as the dextrose 5% control group (40% ± 6%). Although less pronounced, the same trends were observed at 7 days after the treatment. Similarly, SQAd NPs decreased infarct size, as compared to animals receiving free adenosine or dextrose 5% (Figure 6B,C).

The quantification of fibrosis on sections stained with Picrosirius red confirmed these trends, except at 7 days after the treatment (Figure 6D and Appendix A). There was, however, an important variability between mice in each treatment group, which did not allow to reach significance.

Previously, Takahama et al., reported a significant reduction of infarct size in rats after 30 min of ischemia followed by 3 h reperfusion, when receiving an intravenous infusion of PEGylated liposomal adenosine (450 µg/kg/min) for 10 min, 5 min before the onset of reperfusion [15]. Such discrepancy could be explained by the time of analysis (3 h in the Takahama’study versus 3 and 7 days in our study). Nevertheless, both studies agree with the fact that loading adenosine into nanocarriers allows to improve cardiac protection after reperfusion. On the other hand, the present study did not produce a protective effect as those that we have previously observed in a cerebral ischemia/reperfusion model, where NPs accumulated in the pericytes of the vessels and did not cross the blood-brain barrier [17]. In the cardiac ischemia/reperfusion model, a rapid inflammatory process is triggered by the reperfusion, due to the recruitment of inflammatory cells (as PMNs) [29], which results in the appearance of a vascular leakage and intracardiac hemorrhage [30].

### 3.6. Evaluation of Apoptosis on Tissue Sections

Cardiomyocyte apoptosis after ischemia has shown to be one of the major events causing the development of cardiac hypertrophy and heart failure. Due to the anti-apoptotic effect of adenosine, it was hypothesized that SQAd NPs could provide cardioprotection. Thus, myocardial apoptosis was investigated and measured by TUNEL assay on paraffined heart sections (Figure 7). It was observed that SQAd NPs treatment significantly decreased the number of TUNEL-positive nuclei as well as apoptotic cells, after 3 or 7 days of treatment, compared to the group treated with the dextrose 5% control solution. Indeed, the number of apoptotic cells was 2–2.5 times lower than in this control group. At day 3 after treatment, free adenosine induced a decrease in the number of apoptotic cells, similarly to SQAd NPs. However, SQAd NPs were more efficient in decreasing the number of apoptotic cells 7 days post-treatment. This indicates that the encapsulation of adenosine in squalene-based nanoparticles hinders myocardial apoptosis following ischemia/reperfusion injury in a rather prolonged manner, a possible consequence of the protection of adenosine within the NP formulation.

These results could be related to the fact that in the early stages, for up to 3 days, one could have an effect related to a very fast release of adenosine in the blood stream from the NPs. This would explain why the anti-apoptotic effect would be similar to free adenosine at this time point. The effect observed at late times (i.e., 7 days) would be related to a slow and gradual release of adenosine from the NPs. Thus, SQAd NPs could accumulate in the subendothelial space in the area damaged by the ischemia through an EPR effect. From there, the NPs would be captured mainly by endothelial cells and thus adenosine could be released. This hypothesis could explain the anti-apoptotic effect observed at 7 days.

### 3.7. Inflammation Assessment

Besides apoptosis, histological staining to evaluate the impact of SQAd NP treatment on inflammation remains of great interest. At early time points after myocardial ischemia/reperfusion, there is an important recruitment of macrophages in the heart. Initially (1 to 4 days), macrophages M1 with pro-inflammatory capacities are recruited to promote inflammation and tissue destruction [31]. From day 4 to 14 post-MI, M1 macrophages are replaced by M2 macrophages with reparative and anti-inflammatory capacities [32].

To detect the presence of these two kinds of macrophages, we evaluated the number of positive cells for iNOS and CD206, iNOS being a marker for M1 pro-inflammatory macrophages, whereas CD206 is a mannose receptor expressed on M2 macrophages [31,32,33,34].

Our results showed more iNOS positive cells (Figure 8A,C) in the SQAd NPs group than in the Dxt 5% group (*p* < 0.05), but there was no significant difference between SQAd NPs and free Ad groups at day 3 post treatment. Interestingly, at day 7 post-treatment, the number of iNOS positive cells was significantly higher in the SQAd NPs group, as compared to group treated with free adenosine (*p* < 0.005) or Dxt 5% control solution (*p* < 0.05).

The increased number of iNOS positive cells, 3 and 7 days after treatment could be related to the fact that after cell internalization, SQAd NPs act as an intracellular reservoir of adenosine, followed by a sustained release of the nucleoside in the extracellular medium, triggering the activation of A_1_AR [26], resulting in the stimulation of the nuclear factor-κB–specific DNA-protein binding, which in turn initiates the expression of iNOS [35,36].

Using immunohistochemistry, we then assessed the presence of CD206-expressing cells. As shown in Figure 8B,D, at 3 days post-treatment there were more CD206 positive cells in the SQAd NPs group than in the Dxt 5% group (*p* < 0.05) but this difference was no longer present 7 days post treatment, a time-point where no significant difference was found between the different treatments (i.e., Dxt 5%, free Ad and SQAd NPs). These results could be explained by the fact that CD206 positive M2 macrophages are recruited very rapidly to the injured site to induce tissue repair in mice receiving NPs. Deeper analysis, including the investigation of other macrophage markers as well longer-term studies (e.g., after one month), could be of interest to assess tissue changes more precisely. A three days long recruitment of CD206 positive cells to the ischemic site was equally described by Garcia et al., in mice receiving formyl peptide receptor agonist treatment [37].

Other than macrophages, different pro-inflammatory (IL1-α, IL-1β, IL-6, IL-8, IL-13, IL18, TNF-α) or anti-inflammatory (IL-4, IL-10, TGF-β) cytokines are described in the literature as participating in the response on a cardiac level after an ischemic episode [38]. In this sense, some studies have analyzed the expression of certain cytokines in the serum and in the cardiac tissue of animals subjected to ischemia followed or not by reperfusion. Thus, the team of Yang et al. shows that levels of cytokines in serum drop drastically after 24 h of reperfusion in ischemic mice, to levels similar to sham. In infarcted hearts, the levels of analyzed cytokines (IL-2, IL-10, IL12, and IL-17) were significantly decreased at day 1 after MI and remained reduced compared to those in the sham group during the following days. The level of IL-4 in the heart significantly dropped at 3 days after myocardial infarction (MI), and the levels of IL-1β, G-CSF, and TNF-α were remarkably decreased starting at day 5 after MI [39].

Other research groups have observed a return to baseline levels of cytokine mRNA expression within 24 h post-reperfusion in the heart [40,41]. Based on this information, it would have been interesting to analyze cytokines/chemokines in the serum, at early times (up to 24–48 h) and further to compare to the results obtained in the cardiac tissue.

However, it should be noted that some studies underlined that the acute surgical trauma induced by the open chest ischemia/reperfusion model, not only increases background of cytokine induction up to 24 h but also may cause significantly greater variability [42,43].

In our project, we conducted our analysis at day 3 and at day 7 post-reperfusion because the goal was to see if there was a more beneficial effect, in mice that have received SQAd NPs compared to free drug or vehicle. We observed a strong variability between the animals, which is certainly related to the very invasive model.

### 3.8. Assessment of SQAd, Free Adenosine and Adenosine Metabolites on Whole Blood and Heart Tissue

SQAd, free Ad and Ad metabolites (i.e., inosine and hypoxanthine), were determined in blood and heart of rats submitted to cardiac I/R and healthy animals which did not undergo I/R. In healthy rats receiving SQAd nanoparticles, a high blood concentration was observed 5 min after injection, followed by a dramatic decrease at 15 min (Figure 9A).

The blood concentration of adenosine was not significantly modified during 24 h post-injection, compared to its concentration before injection (Figure 9B). Inosine concentrations are relatively high between 5 min and 1 h and return to a basal level at 6 h post injection (Figure 9C).

In animals with cardiac ischemia/reperfusion, blood concentrations of SQAd, adenosine, and inosine decreased between 15 min to 6 h after SQAd intravenous injection, whereas levels of hypoxanthine remained very low (Figure 10). This could be explained by the fact that adenosine is rapidly taken up by erythrocytes and further degraded by plasma adenosine deaminase [7], while inosine and hypoxanthine are more stable adenosine’ metabolites.

Interestingly, the blood concentrations for adenosine, inosine and hypoxanthine were observed as being lower in animals that have undergone ischemia and reperfusion for 6 h, as compared to healthy, untreated control animals (without I/R) (Appendix A).

Subsequently, we quantified SQAd accumulation in cardiac tissue in animals subjected to ischemia and reperfusion for up to 6 h. Thus, we analyzed the ischemic part of the heart and compared it with the non-ischemic part. Our results show a tendency for SQAd accumulation between 15 min and 2 h in the ischemic heart tissue, compared to the non-ischemic area (Figure 11). This could be explained by an enhanced permeability and retention (EPR-like) effect, that would appear after ischemia/reperfusion. Indeed, the EPR effect is due to the presence of inflammatory cells (e.g., macrophages, neutrophils) but also due to the overexpression of vascular endothelial growth factor (VEGF), which lead to an increased vascular permeability [44,45]. Previously, Takahama et al., showed that at 3 h post-reperfusion, adenosine encapsulated in PEGylated liposomes accumulates in ischemic/reperfused myocardium as opposite to free adenosine [15]. In our opinion, more detailed studies would be interesting to better evaluate this EPR-like effect on the heart after I/R. No significant difference was detected between the amounts of adenosine, inosine and hypoxanthine on the ischemic area compared to non-ischemic area (Figure 11). At 6 h, the amount of adenosine, inosine and hypoxanthine were similar in healthy area versus ischemic area in hearts from animals treated with nanoparticles, as well as compared with heart tissue from animals without I/R and without NPs (Appendix A).

This could be explained by the fact that levels of adenosine, inosine and hypoxanthine slow normalize during blood reflow, as previously demonstrated by Hagberg et al., on rats after 15 min of cerebral ischemia [46]. Similarly, Osswald et al., showed on a rat kidney ischemia model that blood recirculation for 15 min after 60 min of ischemia resulted in a decrease of adenosine and inosine levels to values comparable to controls [47].

We note that today there are different strategies to detect and quantify adenosine and its metabolites [48]. However, variability and sensitivity of detection can differ greatly depending on either the analytical method used [17,41] or to the pre-analytical step [49,50].

To our knowledge, we are the first to evaluate levels of adenosine and its metabolites in the blood flow and heart after the administration of adenosine-based nanoparticles in a preclinical cardiac ischemia/reperfusion model.

Our study has some limitations. We have to emphasize that some of the data obtained were not significant from a statistical point of view. Indeed, regarding the model used which is very difficult to obtain, we noted an important variability between the animals. Since the aim was to evaluate the cardiac effect of SQAd NPs on the heart after ischemia/reperfusion, we only included 3 groups in this study (i.e., SQAd NPs, free Ad, and Dxt 5% vehicle). Additionally, we did not administer the NPs as an infusion but as a bolus injection. This choice was made because of the model (it is very difficult to administer an infusion to a 25 g mouse) and considering that the NP formulations enables a continuous circulation of adenosine, and thus, continuous release, which could mimic an infusion.

Of note, after harvesting, the heart tissue was fixed for 24 h in 4% PFA. Unfortunately, this rendered it unusable for any qRTPCR and Western blot studies. These studies could have provided more information concerning the molecular mechanisms established following treatment with SQAd NPs.

## 4. Conclusions

Encapsulating and protecting adenosine from rapid metabolization represents a highly appealing strategy for possible future clinical application. This approach based on covalent bonding between squalene derivative and adenosine results in bioconjugates that self-assemble into nanoparticles. These squalene-adenosine NPs showed anti-platelet aggregation capacities and do not show any toxicity, neither on murine cardiac cell lines, nor on peripheral blood monocellular cells. We underline that squalene is a natural, biocompatible and biodegradable molecule, widely distributed in the human body, which makes this type of nanoparticle very interesting for nanomedicine development. In our study, we did not observe any significant difference in infarct size or area at risk in animals submitted to cardiac ischemia/reperfusion and treated with SQAd NPs, as compared to control animals treated with free adenosine or 5% dextrose solution. However, on a cellular level, a more notable number of apoptotic cells was observed in control groups compared to SQAd NPs treated animals. These results enable the prospect of a more in-depth and longer-term study to better understand the mechanisms established during treatment with these nanomedicines after cardiac reperfusion. For this matter, deeper studies to assess long-term toxicity (over a period of several months) and cardioprotective effects should be carried out in preclinical models closer to humans (e.g., pigs) to overcome the boundaries linked to rodent models.

In conclusion, squalene-based nanotechnology remains appealing to surpass various limitations noted with other types of NPs and could open a new way in nanomedicine field.

## Figures and Tables

**Figure 1 pharmaceutics-15-01790-f001:**
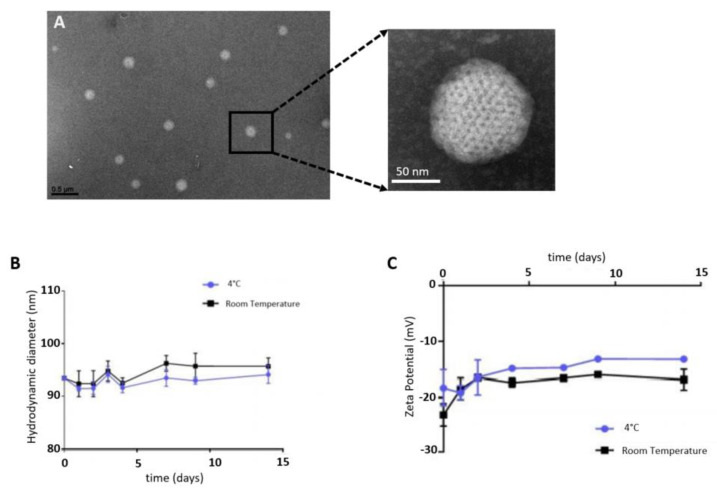
Characterization and stability assessment of SQAd NPs. (**A**) Cryo-TEM images of SQAd NPs obtained after nanoprecipitation. (**B**) DLS monitoring of size and (**C**) zeta potential of SQAd NPs over 14 days in water supplemented with 5% dextrose. In black = stability at room temperature, in blue = stability at 4 °C (*n* = 4 replicates per condition).

**Figure 2 pharmaceutics-15-01790-f002:**
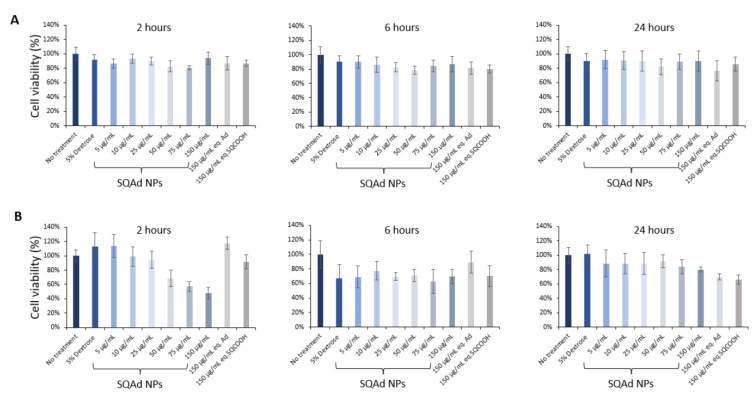
Evaluation of SQAd NPs toxicity. Cell viability was evaluated by MTT colorimetry assay on HL-1 cells (**A**) and H9c2 cells (**B**) after 2 h, 6 h or 24 h of incubation in normoxic conditions (95% air and 5% CO2, 37 °C) with or without treatment. SQAd NPs were tested with increasing concentrations from 5 µg/mL to 150 µg/mL. Free adenosine and SQCOOH NPs (squalenyl acetic acid nanoparticles) were tested as controls at the equivalent maximum dose (*n* = 6 replicates per condition).

**Figure 3 pharmaceutics-15-01790-f003:**
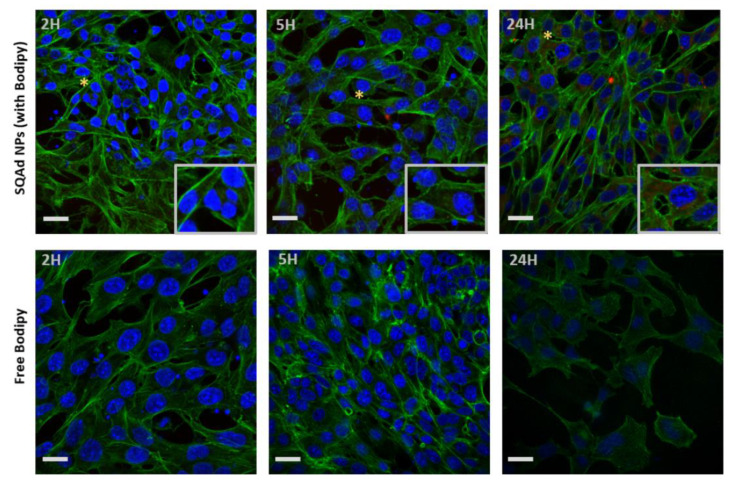
SQAd NPs uptake by HL-1 cardiac cell line. Upper panel, cells incubated with SQAd NPs loaded with Cholesteryl 4,4-difluoro-5-(4-methoxyphenyl)-4-bora-3a,4a-diaza-s-Indacene-3-undecanoate (CholEsteryl BODIPY™, red). The asterisk shows the area with high magnification. Lower panel, cells incubated with only CholEsteryl BODIPY™ (controls). All cells were cultured in normoxic conditions (95% air and 5% CO_2_, 37 °C). After different time-points, cells were rinsed, fixed, and incubated with phalloidin (green) and mounted with medium containing 4′,6-diamidino-2-phenylindole (DAPI, blue). Scale bar = 50 µm.

**Figure 4 pharmaceutics-15-01790-f004:**
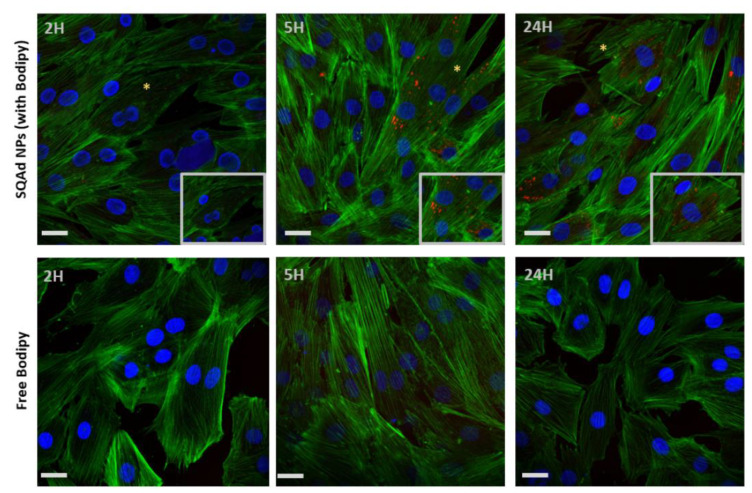
SQAd NPs uptake by H9c2 cardiac cell line. Upper panel, cells incubated with SQAd NPs loaded with Cholesteryl 4,4-difluoro-5-(4-methoxyphenyl)-4-bora-3a,4a-diaza-s-Indacene-3-undecanoate (CholEsteryl BODIPY™, red). The asterisk shows the area with high magnification. Lower panel, cells incubated with only CholEsteryl BODIPY™ (controls). All cells were cultured in normoxic conditions (95% air and 5% CO_2_, 37 °C). After different time-points, cells were rinsed, fixed, and incubated with phalloidin (green) and mounted with medium containing 4′,6-diamidino-2-phenylindole (DAPI, blue). Scale bar = 50 µm.

**Figure 5 pharmaceutics-15-01790-f005:**
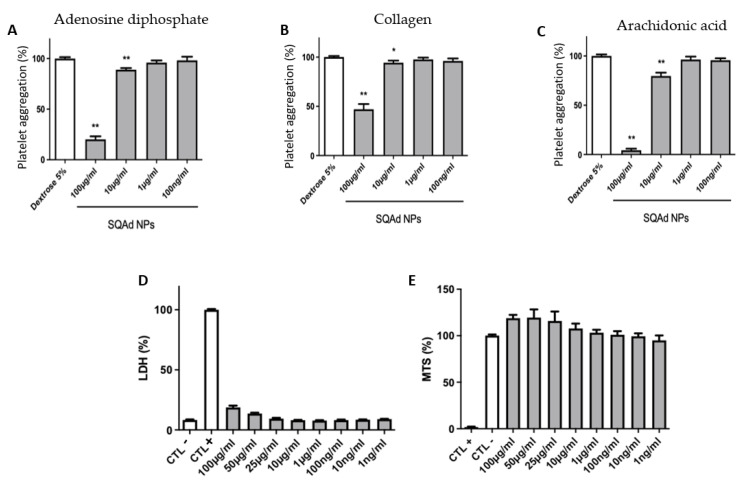
Platelet aggregation and toxicological assessment of SQAd on peripheral blood. Platelet aggregation assessment induced by adenosine diphosphate (**A**), collagen (**B**), or arachidonic acid (**C**), after incubation with different concentration of SQAd NPs. Dextrose 5% is used as a negative control, while Triton 1% is used as positive control. Results are expressed as percentage of response (mean (%) ± SD, *n* = 6). Evaluation of cytotoxicity on PBMC after incubation for 24 h with different concentrations of SQAd NPs by using LDH assay (**D**) or MTS tests (**E**) (mean (%) ± SD, *n* = 3). * *p* < 0.05, and ** *p* < 0.01. Significance is evaluated using GraphPad Prism 7.0 Software with Student *t* test analysis between groups.

**Figure 6 pharmaceutics-15-01790-f006:**
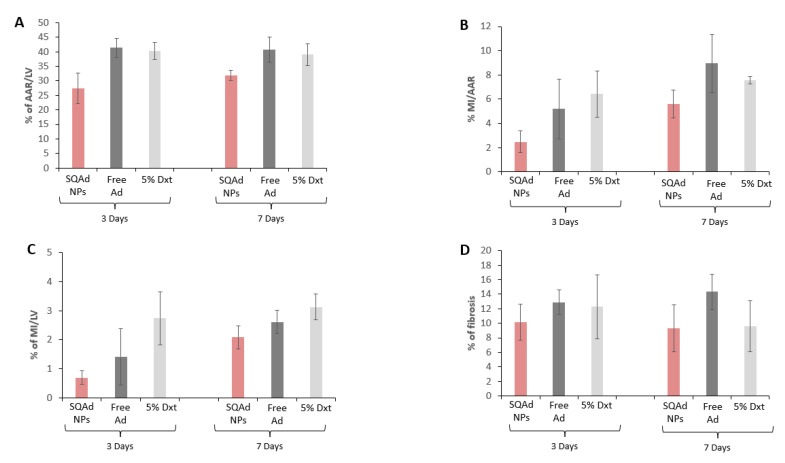
Effect of SQAd NPs treatment on heart after ischemia and reperfusion. (**A**) Area at risk (AAR), (**B**) myocardial infarction (MI), (**C**) percentage of MI/LV and (**D**) fibrosis, assessment on three different groups at 3 or 7 days of reperfusion. Results are expressed as means ± S.E.M of 5–7 animals per condition. The differences between the groups are not significant.

**Figure 7 pharmaceutics-15-01790-f007:**
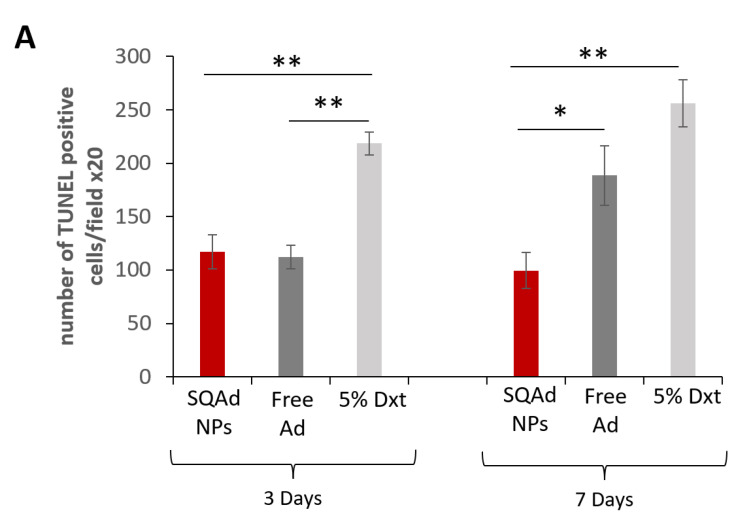
Immunofluorescence evaluation of cell apoptosis on heart sections. (**A**) Graphical comparison of the number of TUNEL-positive nuclei between the different treatment groups. Results are expressed as means ± S.E.M of 5–7 animals per condition. * *p* < 0.05, and ** *p* < 0.01. Significance is evaluated using GraphPad Prism 7.0 Software and Excel Software Statistics (Microsoft Office 365), a Student *t* test analysis was performed between groups. (**B**) Representative images were obtained on heart sections after TUNEL staining, at two different time points (3 and 7 days) and on three groups (SQAd NPs, Free Ad and 5% Dxt). Scale bar 100 µm.

**Figure 8 pharmaceutics-15-01790-f008:**
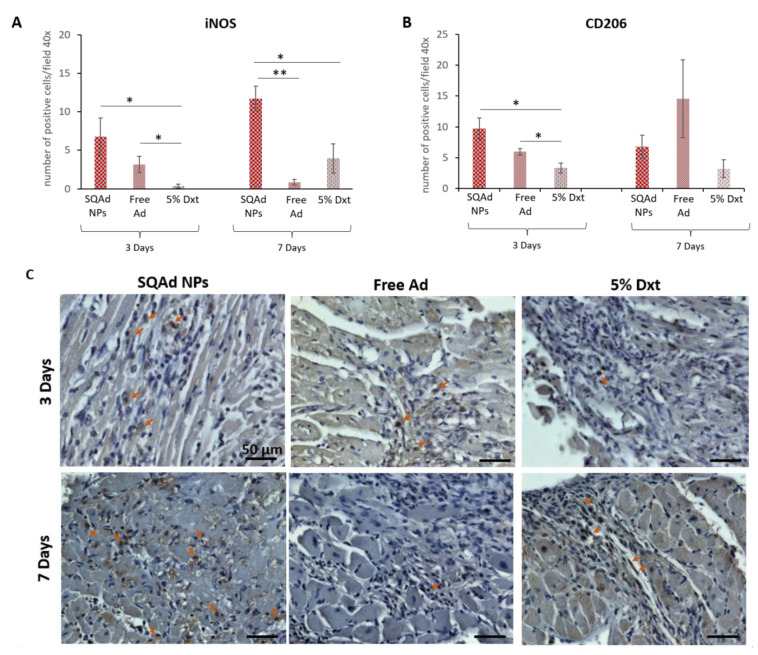
Immunostaining evaluation on heart sections. (**A**,**C**) Positive cells expressing iNOS (Macrophages M1) and (**B**,**D**) positive cells expressing CD206 (Macrophages M2) were assessed. Comparative analysis at three- and seven-days post reperfusion on tissues from animals receiving SQAd NPs, free Ad or 5% Dextrose. * *p* < 0.05, and ** *p* < 0.01. Significance is evaluated using Excel Software Statistics (Microsoft Office 365), a Student *t* test analysis was performed between groups. For (**C**,**D**), the arrows indicate positive cells.

**Figure 9 pharmaceutics-15-01790-f009:**
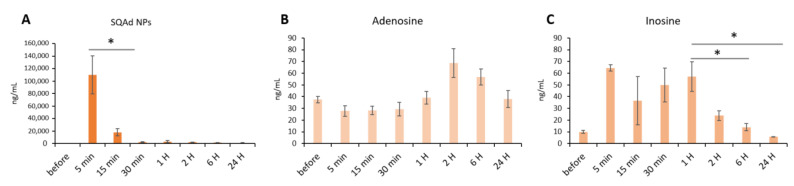
Whole blood quantification in rats without ischemia/reperfusion and receiving SQAd NPs. Quantification of SQAd (**A**), adenosine (**B**) and inosine (**C**), in rats receiving 15 mg/kg SQAd NPs intravenously. * *p* < 0.05. Significance was evaluated using Excel Software Statistics (Microsoft Office 365), and a Student *t* test analysis was performed between groups.

**Figure 10 pharmaceutics-15-01790-f010:**
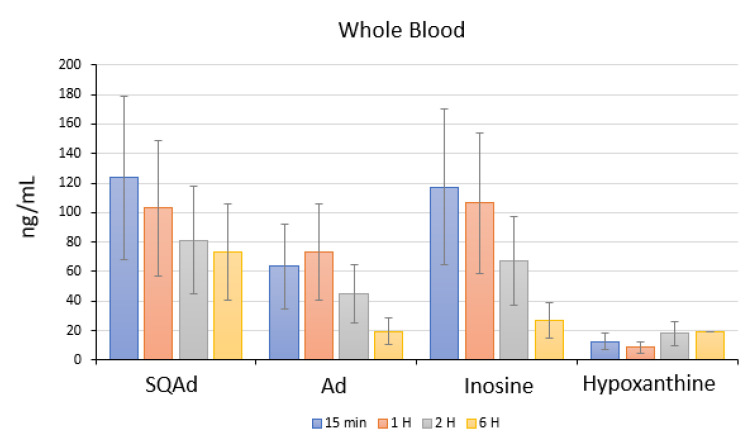
Whole blood quantification in rats with ischemia/reperfusion. Quantification of SQAd, adenosine inosine and Hypoxanthine, in rats receiving 15 mg/kg SQAd NPs intravenously.

**Figure 11 pharmaceutics-15-01790-f011:**
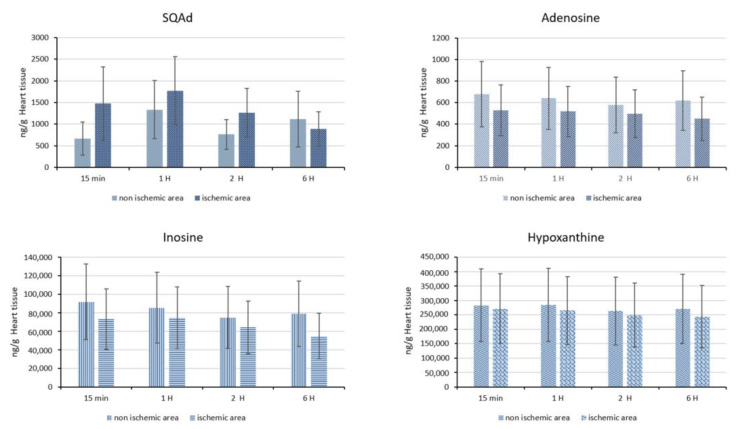
Cardiac quantification. SQAd, adenosine, inosine and hypoxanthine were quantified in rats with ischemia and reperfusion and receiving 15 mg/kg of SQAd NPs intravenously. Ischemic and non-ischemic heart tissues are compared.

## Data Availability

Not applicable.

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
