# Peer review of "Assessment of Squalene-Adenosine Nanoparticles in Two Rodent Models of Cardiac Ischemia-Reperfusion"

_pharmaceutics, 2023, doi:10.3390/pharmaceutics15071790_

Round 1

Reviewer 1 Report (Previous Reviewer 3)

The authors well addressed my concerns and I believe this revised version is more suitable for publication. Besides, similar works are encouraged to be cited: Chin. Chem. Lett. 2023, 34, 107518.; Asian J. Pharm. Sci., 2020, 15, 558-575.

Author Response

Reviewer 1

Comments and Suggestions for Authors

The authors well addressed my concerns and I believe this revised version is more suitable for publication. Besides, similar works are encouraged to be cited: Chin. Chem. Lett. 2023, 34, 107518.; Asian J. Pharm. Sci., 2020, 15, 558-575.”

The authors would like to thank the Reviewer for the time spent reviewing our manuscript and for his/her bibliographical advice.

Following the Reviewer's advice, we have included the reference: Advances of nanoparticles as drug delivery systems for disease diagnosis and treatment, Chin. Chem. Lett. 2023, 34, 107518” (line 80).

Conversely, for the reference “Asian J. Pharm. Sci. 2020, 15, 558-575”, this is an excellent review giving an overview on recent application of cerium oxide nanoparticles, but this is not the field of our research article.

Reviewer 2 Report (New Reviewer)

This study investigates adenosine in reperfusion injury in a preclinical model, using squalene nanoparticles. No toxicity was found. Treatment showed a significant reduction in the diseased and infarcted area, paired with a reduction in apoptosis. Article is well written and deals with an interesting topic. Background, methods, results and discussion are adequate. References are updated. Conclusions are supported by data. Authors should be commended for their work. Reply to previous comments were satisfactory. 

Author Response

The authors would like to thank the Reviewer for the time spent reviewing our manuscript as well as for his/her very positive comments.

Reviewer 3 Report (New Reviewer)

In the manuscript, the authors describe “Assessment of Squalene-Adenosine nanoparticles in two rodent models of cardiac ischemia-reperfusion. Researchers suggested that encapsulate adenosine in squalene-based nanoparticles has biologically informative and cardioprotective which will be providing new therapeutic strategies.

This is a very interesting paper that introduces very important information about the SQAd NPs.

Conclusions may be more appealing if authors include key insights and future prospects, rather than just summarizing.

Author Response

The authors would like to thank the Reviewer for the time spent reviewing our manuscript. Following the Reviewer's advice, we have modified the Conclusion section as following:

“ 4. Conclusions

Encapsulating and protecting adenosine from rapid metabolization represents a highly appealing strategy for possible future clinical application. This approach based on covalent bonding between squalene and adenosine results in a bioconjugate that self-assembles into nanoparticles. These squalene-adenosine NPs showed anti-platelet aggregation capacities and do not show any toxicity, neither on murine cardiac lines, nor on peripheral blood monocellular cells. We underline that squalene is a natural, biocompatible and biodegradable molecule, widely distributed in the human body, which makes this type of nanoparticle very interesting for nanomedicine development. In our study, we did not observe any significant difference in infarct size or area at risk in animals submitted to cardiac ischemia/reperfusion and treated with SQAd NPs, as compared to control animals treated with free adenosine or 5% dextrose solution.  However, on a cellular level, a more notable number of apoptotic cells was observed in control groups compared to SQAd NPs treated animals. These results enable the prospect of a more in-depth and longer-term study to better understand the mechanisms put in place during treatment with these nanomedicines after cardiac reperfusion. For this matter, deeper studies to assess long-term toxicity (over a period of several months) and cardioprotective effects should be carried out in preclinical models more similar to humans (e.g. pigs) to overcome the boundaries linked to rodent models.

In conclusion, squalene-based nanotechnology remains appealing to surpass various limitations noted with other types of NPs and could open a new way in nanomedicine field.”

This manuscript is a resubmission of an earlier submission. The following is a list of the peer review reports and author responses from that submission.

Round 1

Reviewer 1 Report

In their manuscript, Brusini et al. investigated the Squalene-Adenosine nanoparticles in rodent models (mouse and rat) of cardiac ischemia-reperfusion.

The study has some limitations that were underlined by the authors and the research does not investigate in-depth the effects of SQAd nanoparticles in cardiac I/R treatment. Therefore, caution is needed when the protective effect on I/R is claimed.

The manuscript needs improvements to be worth being published.

1) The title “Assessment of Squalene-Adenosine nanoparticles in a rodent model of cardiac ischemia-reperfusion” should be amended. In fact, the SQAd NPs assessment is done in two animal models (mouse and rat), a mouse model to follow the therapeutic effects and a rat model to follow the pharmacokinetics and heart accumulation of NPs.

2) Figure 3. The cellular uptake of SQAd NPs is barely visible. The image contrast should be improved. How these NPs are internalized by the cardiac cells? Did the authors perform competitive studies? Previously, the same group showed that LDLR is involved in SQAd NPs internalization by HepG2 cells (DOI: https://doi.org/10.1124/jpet.118.254961).  Is the same pathway involved in cardiomyocytes?

3) The hemolytic effect was followed in whole blood (section 3.4.) or using washed erythrocytes (section 2.6)

4) Figure 5. Please, provide representative images of Evans blue and Picrosirius red staining (above the graphs). Show statistical significance on the graphs, if there is any.

5) Lines 482-485. The authors cannot claim a “protective effect of SQAd NPs” since no statistical significance is obtained. Revise it.

6) Revise lines 496-499. It seems that something is missing.

7) Figure 6 and Figure 7. Provide the representative images supporting the graphs.

8) Please provide some details regarding the assessment of SqAd, Ad, Ad metabolites by Quality Assistance S.A.

9) Figure 8. -Statistical significance?

10) Lines 652-653. Titles for Figures S1 and S2: Whole blood/ cardiac quantification…… add “ of adenosine, inosine, and hypoxanthine” at 6 hours “after……”-indicate the experimental condition.

Author Response

Author's Reply to the Reviewer 1

Comments and Suggestions for Authors

In their manuscript, Brusini et al. investigated the Squalene-Adenosine nanoparticles in rodent models (mouse and rat) of cardiac ischemia-reperfusion.

The study has some limitations that were underlined by the authors and the research does not investigate in-depth the effects of SQAd nanoparticles in cardiac I/R treatment. Therefore, caution is needed when the protective effect on I/R is claimed.

The manuscript needs improvements to be worth being published.

  • The title “Assessment of Squalene-Adenosine nanoparticles in a rodent model of cardiac ischemia-reperfusion” should be amended. In fact, the SQAd NPs assessment is done in two animal models (mouse and rat), a mouse model to follow the therapeutic effects and a rat model to follow the pharmacokinetics and heart accumulation of NPs.

Response: The authors thank the Reviewer for reviewing our manuscript and for his/her comments that gives us the opportunity to improve it.

As pointed out in the first comment, two rodent models were indeed used in this study.
As a result, we have modified the title of the article as follows:
“Assessment of Squalene-Adenosine nanoparticles in two rodent models of cardiac ischemia-reperfusion”

  • Figure 3. The cellular uptake of SQAd NPs is barely visible. The image contrast should be improved. How these NPs are internalized by the cardiac cells? Did the authors perform competitive studies? Previously, the same group showed that LDLR is involved in SQAd NPs internalization by HepG2 cells (DOI: https://doi.org/10.1124/jpet.118.254961).  Is the same pathway involved in cardiomyocytes?

Response: To follow the Reviewer’s comment, the contrast in Figure 3 has been improved, and an insert with high magnification has been included.

       Figure 3: SQAd NPs uptake by HL1 cardiac cell line. Upper panel, cells incubated with SQAd NPs loaded with Cholesteryl 4,4-difluoro-5-(4-methoxyphenyl)-4-bora-3a,4a-diaza-s-Indacene-3-undecanoate (CholEsteryl BODIPY™, red). The asterisk shows the area with high magnification. Lower panel, cells incubated with only CholEsteryl BODIPY™ (controls). All cells were cultured in normoxic conditions (95 % air and 5 % CO2, 37°C). After different time-points, cells were rinsed, fixed, and incubated with phalloidin (green) and mounted with medium containing 4’,6-diamidino-2-phenylindole (DAPI, blue). Scale bar = 50 µm.

Indeed, our group has previously demonstrated that LDLR is involved in SQAd NPs internalization by HepG2 cells, cited as reference 33 (doi: 10.1124/jpet.118.254961.) in the manuscript. As described by Rouquette et al., the HepG2 cell line was chosen because SQAd NPs accumulate in liver tissue after intravenous administration, another result observed by our group (doi: 10.1016/j.jconrel.2015.06.016). An internalization study on cardiac cells is underway to determine by which mechanisms NPs are captured by the cardiac cells; these results will be obtained in several additional months.

  • The hemolytic effect was followed in whole blood (section 3.4.) or using washed erythrocytes (section 2.6)

Response: Thank you for this comment. The experiments were done on whole blood. To avoid confusion, the section 2.6 has been modified (please see lines 196-209).

  • Figure 5. Please, provide representative images of Evans blue and Picrosirius red staining (above the graphs). Show statistical significance on the graphs, if there is any.

Response: Thank you for this comment. No statistical significance was observed for fibrotic area, myocardial infarction, or area at risk. This precision has been added to the legend of Figure 5 (lines 508-509). Further, we have added a supplementary figure (Supplementary Figure 1) with Picrosirius Red staining.

Supplementary Figure S1: Picrosirius red staining revealed the presence of fibrosis on heart sections obtained from mice submitted to 30 minutes of ischemia and followed by 3- or 7-days of reperfusion. The inset shows the magnification of a selected area. Representative images of picrosirius red staining (2.5×; insets, 10×) for each group.

5) Lines 482-485. The authors cannot claim a “protective effect of SQAd NPs” since no statistical significance is obtained. Revise it.

Response: The phrase “Overall, these data confirmed the protective effect of SQAd NPs.” has been deleted (line 502).

6) Revise lines 496-499. It seems that something is missing.

Response : Thank you for this remark, the error has been corrected as follows: “On the other hand, the present study did not produce a protective effect as those that we have previously observed in a cerebral ischemia/reperfusion model, where NPs accumulated in the pericytes of the vessels and did not cross the blood-brain barrier [16]. » (line 517).

7) Figure 6 and Figure 7. Provide the representative images supporting the graphs.

Response: As requested by the Reviewer, we have added representative images next to the graphics.

Figure 6: Immunofluorescence evaluation of cell apoptosis on heart sections.  (A) Graphical comparison of the number of TUNEL-positive nuclei between the different treatment groups. Results are expressed as means ± S.E.M of 5-7 animals per condition. *P < 0.05, and **P < 0.01. Significance is evaluated using GraphPad Prism 7.0 Software and Excel Software Statistics, a Student t test analysis was performed between groups. (B) Representative images were obtained on heart sections after TUNEL staining, at two different time points (3 and 7 days) and on three groups (SQAd NPs, Free Ad and 5% Dxt). Scale bar 100 µm.

Figure 7: Immunohistochemistry evaluation. (A & C) Positive cells expressing iNOS (Macrophages M1) and (B & D) positive cells expressing CD206 (Macrophages M2) were assessed. Comparative analysis at three- and seven-days post reperfusion on tissues from animals receiving SQAd NPs, free Ad or 5% Dextrose. *P < 0.05, and **P < 0.01. Significance is evaluated using Excel Software Statistics, a Student t test analysis was performed between groups. The arrows indicate positive cells. For (D), the asterisk shows the area with high magnification.

8) Please provide some details regarding the assessment of SqAd, Ad, Ad metabolites by Quality Assistance S.A.

Response. Following the Reviewer’s suggestion we have added some precision concerning the protocol developed by Quality Assistance as follows:  “The procedure developed involves a protein precipitation step and separation by UPLC under reverse phase conditions. The different products were detected by tandem MS (Mass Spectrometry). The limit of the detection was 1 to 500 ng/mL for Adenosine, 5 to 250 ng/mL for Hypoxanthine, 5 to 250 ng/mL for Inosine and 5 to 200 ng/mL for SqAd. “ (Please see lines  375-379).

9) Figure 8. -Statistical significance?

 Response: statistical significance was added (please, see figure 8 A and figure 8 C).

Figure 8: Whole blood quantification of SQAd (A), adenosine (B) and inosine (C) in rats without ischemia/reperfusion, receiving SQAd NPs (15mg/kg) intravenously.   *P < 0.05. Significance is evaluated using Excel Software Statistics, a Student t test analysis was performed between groups.

10) Lines 652-653. Titles for Figures S1 and S2: Whole blood/ cardiac quantification…… add “ of adenosine, inosine, and hypoxanthine” at 6 hours “after……”-indicate the experimental condition.

Response: The authors thank for this correction. The titles  of the Figures (now (Supplementary Figure S2 and  Figure S3) were modified as follows:

“Supplementary Figure S2: Whole blood quantification of adenosine, inosine and hypoxanthine. 250 µL of whole blood was taken from rats submitted to ischemia (30 min) and reperfusion (6 hours) and without prior administration of SQAd NPs. As control, 250 µL of whole blood was taken from healthy rats without prior administration of SQAd NPs.”

“Supplementary Figure 3: Cardiac quantification of adenosine, inosine and hypoxanthine.

Quantification was done on heart tissues from rats without or with ischemia (30 min) and reperfusion (6 hours), without prior administration of SQAd NPs. In rats with ischemia and reperfusion a comparison between ischemic and non-ischemic zone was performed.”

Reviewer 2 Report

Well written and presented, although main issues - all centered around the lack of statistcal significance in vivo - are evident even if properly addressed and commented upon.

Author Response

Comments and Suggestions for Authors

Well written and presented, although main issues - all centered around the lack of statistcal significance in vivo - are evident even if properly addressed and commented upon.

Response: The authors would like to thank the Reviewer for the time spent reviewing this article and for his/her very positive comment.

Reviewer 3 Report

In this study, Brusini et al proposed an adenosine-loaded squalene-based nanoparticles (SQAd NPs) and evaluated the cardioprotective role of SQAd NPs in preclinical cardiac ischemia. The data is abundant, however, additional clarification and data are required to supply before publication in this journal.

1.     Adenosine is hydrophilic, it’s hard to be encapsulated inside the hydrophobic core of squalene-based nanoparticles. I assume adenosine is located outside of squalene-based nanoparticles via covalent linker. So, I’m wondering if the adenosine on the surface of SQAd NPs still have the chance to be metabolized during circulation, please clarify.

2.     How does adenosine release from SQAd NPs? Did the authors tested the release profile of adenosine?

3.     SQAd NPs may improve the pharmacokinetic profile of adenosine, however, whether SQAd NPs could target myocardial ischemia remain questionable, please provide solid data to prove it.

4.     After treatment with SQAd NPs, both M1 and M2 macrophages showed an increased percentage. It’s hard to determine the anti-inflammation effect of SQAd NPs. I would suggest testing the cytokines release to support their hypothesis.

Author Response

Comments and Suggestions for Authors

In this study, Brusini et al proposed an adenosine-loaded squalene-based nanoparticles (SQAd NPs) and evaluated the cardioprotective role of SQAd NPs in preclinical cardiac ischemia. The data is abundant, however, additional clarification and data are required to supply before publication in this journal.

  1. Adenosine is hydrophilic, it’s hard to be encapsulated inside the hydrophobic core of squalene-based nanoparticles. I assume adenosine is located outside of squalene-based nanoparticles via covalent linker. So, I’m wondering if the adenosine on the surface of SQAd NPs still have the chance to be metabolized during circulation, please clarify.

Response: The authors thank the Reviewer for this excellent question. 

In SQAd NPs, a fraction of adenosine attached to squalene (SQAd bioconjugate) is exposed to blood at the surface of the NPs but is also located inside the NPs. Indeed, the internal structure of SQAd NPs was investigated by Small-Angle X-ray Scattering (SAXS) and an inverse two-dimensional hexagonal phase was evidenced, resulting from the stacking of inverse cylinders constituted of adenosine-squalene bioconjugates (10.1016/j.jconrel.2019.06.040). Even if a fraction of bioconjugate is located at the surface of the NPs, with the adenosine moiety exposed to the enzymes of the blood, it will not be metabolized as long it remains attached to squalene (which provides protection from metabolization). Indeed, the squalene moiety was conjugated to the amino group of the nucleobase of adenosine which is sensitive to deamination by adenosine deaminase enzymes. Thus, the presence of squalene prevents SQAd metabolization. Moreover, it was highlighted that the prodrug starts to be metabolized once released from the nanostructure (see doi: 10.1016/j.jconrel.2015.06.016, doi: 10.1124/jpet.118.254961 ).

A clarification was made in the text (see lines 406-410).

  1. How does adenosine release from SQAd NPs? Did the authors tested the release profile of adenosine?

Response: The authors thank the Reviewer for this interesting question.  Previously, we have already demonstrated that blood proteins such as low-density lipoproteins (LDLs) interact with SQAd NPs after intravenous administration, resulting in the insertion of the SQAd molecules into LDL. Then, the SQAd bioconjugates loaded into LDL were able to recognize cells expressing the LDL receptors before internalization via clathrin-dependent endocytosis. Afterwards, SQAd underwent a slow enzymatic activation under acidic conditions in the intracellular endo-lysosomal compartments, allowing the production of free adenosine. The adenosine was found to be finally released in the extracellular medium (doi.org/10.1124/jpet.118.254961). The release profile of Ad from SQAd NPs was studied by incubating the NPs in an acidic environment mimicking the lysosomal compartment of the cells (pH 4.8, 37°C). HPLC analysis indicated that up to 20% of SQAd was degraded after 24 hours of incubation, leading to the release of free adenosine (retention time 3.9 minutes) (doi.org/10.1124/jpet.118.254961).

  1. SQAd NPs may improve the pharmacokinetic profile of adenosine, however, whether SQAd NPs could target myocardial ischemia remain questionable, please provide solid data to prove it.

Response: This is a very good remark. Previously, different papers (DOI: 10.1016/j.jconrel.2003.10.008; DOI: 10.1002/cmmi.1501) reported an “Enhanced Permeability and Retention” (EPR) like effect in the heart after ischemia. The EPR effect is due to the presence of inflammatory cells (e.g. macrophages, neutrophils) but also due to the overexpression of vascular endothelial growth factor (VEGF), which leads to an increased vascular permeability.  We believe that this mechanism could explain the accumulation of the nanoparticles in the heart. This has been clarified in the text (please see lines 626-632).

In the future, we will consider to deeper investigate the mechanisms of accumulation of squalene-based nanoparticles into the heart after cardiac ischemia and reperfusion. 

  1. After treatment with SQAd NPs, both M1 and M2 macrophages showed an increased percentage. It’s hard to determine the anti-inflammation effect of SQAd NPs. I would suggest testing the cytokines release to support their hypothesis.

Response: The reviewer is right. Unfortunately, we do not have enough serum from these mice to use for further studies.

Round 2

Reviewer 1 Report

The authors considered the comments and mostly answered.

- Improve the quality of Figure 6B. From the images provided it seems that for SqAd NPs, the % Tunel-positive cells (red, barely visible) is higher at 7 days compared to 3 days not in agreement with the graph. 

- In Figure 7, images provided in part C depict immunohistochemistry for iNOS and in part D, the immunofluorescence for CD 206. Also, line 564 refers to immunofluorescence instead immunohistochemistry. The resolution of the images in Figure 7D should be improved. 

Reviewer 3 Report

I don't believe the authors addressed my concerns well, necessary experiments are required to supply to make this study more solid.